# Spoken Question Answering and Speech Continuation using Spectrogram-Powered LLM

**Eliya Nachmani**[1,*], **Alon Levkovitch**[1,3,*,†], **Roy Hirsch**[2], **Julian Salazar**[1],
**Chulayuth Asawaroengchai**[1], **Soroosh Mariooryad**[1], **Ehud Rivlin**[2],
**RJ Skerry-Ryan**[1], **Michelle Tadmor Ramanovich**[1]
[1]Google Research, [2]Verily AI, [3]Tel-Aviv Univeristy
{eliyn, alevkovitch, royhirsch}@google.com

## ABSTRACT

We present Spectron, a novel approach to adapting pre-trained large language models (LLMs) to perform spoken question answering (QA) and speech continuation. By endowing the LLM with a pre-trained speech encoder, our model becomes able to take speech inputs and generate speech outputs. The entire system is trained end-to-end and operates directly on spectrograms, simplifying our architecture. Key to our approach is a training objective that jointly supervises speech recognition, text continuation, and speech synthesis using only paired speech-text pairs, enabling a 'cross-modal' chain-of-thought within a single decoding pass. Our method surpasses existing spoken language models in speaker preservation and semantic coherence. Furthermore, the proposed model improves upon direct initialization in retaining the knowledge of the original LLM as demonstrated through spoken QA datasets. We release our audio samples and spoken QA dataset via our website.[1]

## 1 INTRODUCTION

The goal of natural language processing (NLP) is to develop computational models that can understand and generate human language. By capturing the statistical patterns and structures of text-based natural language, language models can predict and generate coherent and meaningful sequences of words. Combined with the Transformer model architecture (Vaswani et al., 2017), large language models (LLMs) trained on web-scale amounts of text, with proportionate compute and size, have demonstrated remarkable success in NLP tasks (Devlin et al., 2019; Brown et al., 2020; Chowdhery et al., 2022; Zhang et al., 2022a; Scao et al., 2022; Zeng et al., 2023). However, transferring these abilities to *spoken* human language remains a challenging frontier. Spoken dialog systems remain a cascade of separately trained automatic speech recognition (ASR), natural language understanding (NLU) and generation (NLG), and text-to-speech (TTS) systems (Gorin et al., 1997; Jokinen & McTear, 2009), with LLMs now playing the role of a combined NLU and NLG system. However, such cascades introduce latency and additional mechanisms for propagating and rendering non-verbal cues like speaker identity and prosody. Recently, spoken language models (Lakhotia et al., 2021; Kharitonov et al., 2022) and other generative audio models (Dhariwal et al., 2020; Hawthorne et al., 2022; Borsos et al., 2023; Agostinelli et al., 2023) have emerged as a promising avenue for generative speech modeling. These works quantize audio representations (Hsu et al., 2021; Chung et al., 2021; Zeghidour et al., 2022; Défossez et al., 2022) into learned discrete tokens compatible with the same next-token cross-entropy objective as text LLMs, a step that (Nguyen et al., 2022) argued as necessary for generative quality. In this paper, we introduce Spectron, a novel spoken language model that:

- Directly process spectrograms as both input and output. Spectron leverages the audio capabilities of a pre-trained speech encoder through the use of intermediate projection layers.

---

*Equal contribution.

†Work done during an internship at Google.

[1]https://michelleramanovich.github.io/spectron/spectron

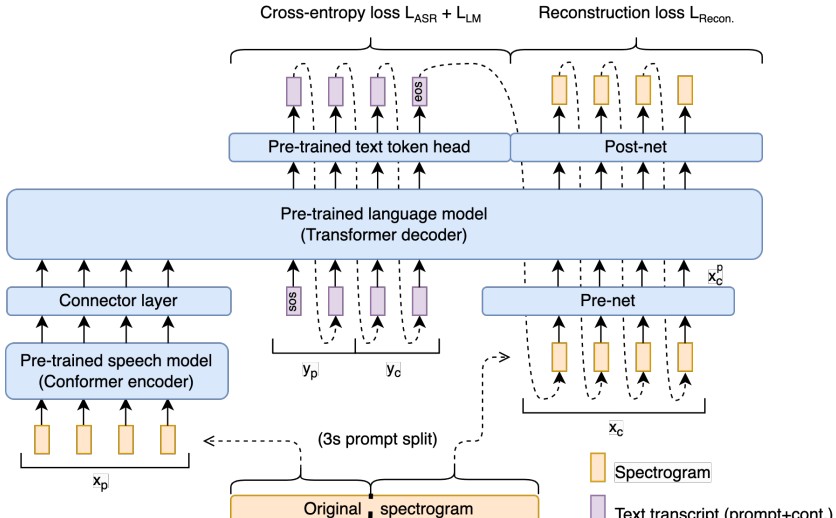

Figure 1: Spectron connects the encoder of a speech recognition model with a pre-trained Transformer decoder language model. At training time, we take speech utterances and split their audio into a *prompt* and its *continuation*. From the prompt speech features, the full (prompt and continuation's) transcript must be reconstructed, as well as the continuation's speech features via newly introduced pre- and post-net speech modules. At inference time, only a prompt is provided; the prompt's transcription, text continuation, and speech continuations are all generated by the model.

- Demonstrably transfer generative ability from a pre-trained LLM, as shown by competitive performance in semantic coherence and spoken question answering over other end-to-end spoken language models.

To quantify this transfer of knowledge, we also introduce two benchmarks for the nascent spoken QA task, which we synthesize from Web Questions (Berant et al., 2013) and generations from LLaMA (Touvron et al., 2023). Audio samples and our LLaMA dataset can be found on the project website given on the first page.

Our work shows that the inductive biases from a pre-trained speech encoder and a language model decoder enable end-to-end training and state-of-the-art performance without sacrificing representational fidelity. Key to this is a novel end-to-end training objective which implicitly supervises speech recognition, text continuation, and conditional speech synthesis in a joint manner. The language model transcribes and generates text continuations, acting as an 'intermediate scratchpad' (Nye et al., 2021; Wei et al., 2022) to be conditioned on for audio generation. A novel spectrogram regression loss also supervises the model to match the higher-order temporal and feature deltas of the ground truth, based on the idea that the derivatives of the ground truth express rich, longer-range information about the shape of the signal. Our overall scheme is summarized in Figure 1 and described in the rest of this work.

## 2 RELATED WORK

The dominant approach to spoken language modeling is to use compact discrete speech representations. This allows the application of text-based language models to speech data. Typically, these representations are created by clustering the outputs of a speech encoder using K-means and taking the centroids as tokens. The resulting discrete sequences can be easily modeled using Transformer architectures (Vaswani et al., 2017). Below are notable examples of works using this approach; a comparison table is also presented in Appendix A.2.

**Generative Spoken Language Modeling** (GSLM; Lakhotia et al., 2021) offers a baseline system that operate on units quantized from pre-trained audio representations, such as HuBERT (Hsu et al., 2021). The quantized units are processed by a Transformer-based model. An additional unit-to-speech

decoder converts the generated units into spectrograms. Spectron's approach is more simple and explicit, where a single model is input with spectrograms and outputs raw spectrograms.

**TWIST** (Hassid et al., 2023) uses the same unit-to-speech and speech-to-unit systems as GSLM, but warm-starts the spoken language model from a text-based language model. They show that this warm-start improves overall metrics and convergence speed, with reasonable performance on StoryCloze tasks (though notably degraded from the text model). For the textual language model, they use the state-of-the-art and open-weight OPT (**?**) and LLaMA models up to 7B (13B at camera-ready time) and show that spoken language model improvements scale.

**AudioLM** (Borsos et al., 2023) utilizes two kinds of quantized representations: w2v-BERT (Chung et al., 2021) as semantic tokens, and SoundStream (Zeghidour et al., 2022) as acoustic tokens. SoundStream embeddings undergo discretization using residual vector quantization (RVQ), resulting in a hierarchy of vector quantizers. AudioLM utilizes three transformer models, each corresponding to a different layer of token generation. As Spectron does not involve any quantization, our method naturally preserves the input's semantic and acoustic characteristics. Furthermore, Spectron offers a single model trained with a unified objective, as opposed to the multiple components of AudioLM.

**SpeechGPT** (Zhang et al., 2023a) adapts LLaMA-7B to perform speech tasks by using both discrete speech representations and text. They introduce the SpeechInstruct dataset which they use for instruction tuning. SpeechGPT is trained in 3 different steps: modality adaptation, cross-modal instruction fine-tuning, and chain-of-modality instruction fine-tuning (using LoRA; Hu et al., 2022). The obtained model is capable of generating both speech and text, as well as following instructions in both modalities. Our method, in comparison, is trained using a single reconstruction step and uses only public datasets for integration. Despite not using a curated dataset or specialized prompts, we demonstrate competitive performance on spoken question answering and superior results for speech continuation.

As for related works in other tasks: **Spoken language understanding (SLU)**: A number of recent studies explore the usage of pre-trained language models (LMs) for different SLU tasks. Gong et al. (2023), Zhao et al. (2023), and Liu et al. (2023a) fine-tuned LMs on audio data to perform speech-to-text question answering tasks, that is, answer textual questions about directly-input audio. Fathullah et al. (2023) showed that adding an audio encoder to an LM and training with LoRA enables the LM to perform automatic speech recognition (ASR). Zhang et al. (2022b) aligned text and audio tokens to perform a large number of SLU tasks. Peng et al. (2023) showed that LMs can be used to answer text questions about spoken properties of language. Spectron, in comparison, produces both textual outputs and spectrograms using a single autoregressive decoder. **Multi-modal text-speech training**: Ao et al. (2022) performed joint training on speech and text data to perform multiple tasks, such as text-to-speech (TTS) and ASR. Ren et al. (2019) used unsupervised pre-training on text and speech data to perform TTS for low-resource languages. **Textual-guided audio generation**: Liu et al. (2023b) used LMs to generate audio scripts and interacting with audio-creation APIs. Huang et al. (2023) augmented the ChatGPT input/output interface to invoke ASR / TTS tools.

## 3 APPROACH

### 3.1 ARCHITECTURE

We propose a novel architecture for direct speech continuation. The architecture is initialized with a pre-trained speech encoder denoted as $\mathcal{E}$ and a pre-trained language decoder denoted as LM. The encoder is prompted with a speech utterance as input, which it encodes into continuous linguistic features. These features are fed into the decoder as a prefix, and the whole encoder-decoder is optimized to jointly minimize a cross-entropy loss (for speech recognition and transcript continuation) and a novel reconstruction loss (for speech continuation). During inference, one provides a spoken speech prompt, which is encoded and then decoded to give both text and speech continuations.

#### 3.1.1 INPUT PRE-PROCESSING

During training, the proposed model uses supervised speech utterances, which are pairs of speech $x$ and transcripts $y$ for training. The speech input, denoted as $x$, is a spectrogram that is split into two

segments at position $s$:

$$x_p = x_{\leq s}, \quad x_c = x_{>s}. \tag{1}$$

The first segment $x_p$ (which we call the *prompt*) is fed into the speech encoder $\mathcal{E}$ to give continuous representations that condition the LM. The second segment $x_c$ (the *continuation*) is used later for a spectrogram reconstruction loss. SpecAugment (Park et al., 2019) is applied for data augmentation. The corresponding transcripts $y$ can be also split at position $\phi(s)$:

$$y_p = y_{\leq \phi(s)}, \quad y_c = y_{>\phi(s)}, \tag{2}$$

where $\phi(s)$ maps the feature index $s$ in $x$ to its text token index in $y$. Note that $\phi(s)$ is not needed for our training losses.

### 3.1.2 SPEECH ENCODER

The speech encoder $\mathcal{E}$ is a 600M-parameter Conformer encoder (Gulati et al., 2020) pre-trained on web-scale data (12M hours; Zhang et al., 2023b). It takes the spectrogram of the source speech as input, generating a hidden representation that incorporates both linguistic and acoustic information. The input spectrogram is first subsampled using a convolutional layer and then processed by a series of Conformer blocks. Each Conformer block consists of a feed-forward layer, a self-attention layer, a convolution layer, and a second feed-forward layer. The outputs of the total encoder $\mathcal{E}$ are passed through a layer $\mathcal{P}$ that projects the hidden representations into the embedding dimension of the language model. We denote these final embeddings

$$x_p^{\text{lm}} = \mathcal{P}(\mathcal{E}(x_p)). \tag{3}$$

### 3.1.3 LANGUAGE MODEL

We use prefix decoder language models with 350M or 1B parameters trained in the manner of PaLM 2 (Google, 2023), which we denote as $\text{LM}(-)$. The LM receives the encoded features of the prompt $x_p^{\text{lm}}$ as a prefix. Note that this is the only connection between the speech encoder and the LM decoder; i.e., there is no cross-attention between the encoder and the decoder. This late-stage integration is consistent with work in ASR, which found that joint fine-tuning of a pre-trained speech encoder and a pre-trained LM decoder into a sequence-to-sequence model can improve performance, even if the integration occurs as a single final layer (Deng et al., 2021); more layers did not improve performance, which they attribute to having sufficiently powerful text representations. During training, the decoder is teacher-forced to predict the text transcription $y_p$, text continuation $y_c$, and speech embeddings $x_c^p$. To convert the speech embeddings to and from spectrograms, we introduce lightweight modules $h^{\text{pre}}$ and $h^{\text{post}}$, described in the next section. In all, we get next-step predictions for the concatenation of these text tokens and embeddings:

$$[\hat{y}_p, \hat{y}_c, \hat{x}_c^p] = \text{LM}(x_p^{\text{lm}}, [y_p, y_c, x_c^p]). \tag{4}$$

By having the *same* architecture decode the intermediate text and the spectrograms, we gain two benefits. First, we benefit from the pre-training of the LM in the text domain to continue the prompt in the text domain before synthesizing the speech. Secondly, the predicted text serves as intermediate reasoning, enhancing the quality of the synthesized speech, analogous to improvements in text-based language models when using intermediate scratchpads (Nye et al., 2021) or chain-of-thought (CoT; Wei et al., 2022).

### 3.1.4 ACOUSTIC PROJECTION LAYERS

To enable the language model decoder to model speech features, we employ a multi-layer perceptron (MLP), the *pre-net* $h^{\text{pre}}$ to project the ground truth spectrogram speech continuations $x_c$ to the language model dimension $x_c^p = h^{\text{pre}}(x_c)$. This pre-net $h^{\text{pre}}$ compresses the spectrogram input $x_c$ into a lower dimension, creating a bottleneck that aids the decoding process. This bottleneck mechanism prevents the model from repetitively generating the same prediction in the decoding process, as demonstrated in previous work (Shen et al., 2018). To project $\hat{x}_c^p$ from the language model dimension to the spectrogram dimension, the model employs a *post-net* $h^{\text{post}}$, which is also an MLP. This projection is represented by $\hat{x}_c = h^{\text{post}}(\hat{x}_c^p)$.

Both $h^{\text{pre}}$ and $h^{\text{post}}$ are two-layer MLPs. Additionally, the input text sequence $[y_p, y_c]$ is padded at the beginning with a "start of sequence" (sos) token, while the output sequence is padded with an "end of sequence" (eos) token at the final position.

### 3.2 TRAINING OBJECTIVE

The training methodology of the proposed approach is depicted in Figure 1. It uses two distinct loss functions: (1) cross-entropy loss, employed for both speech recognition and transcript continuation, and (2) regression loss, employed for speech continuation. During training, all parameters are updated (speech encoder $\mathcal{E}$, projection layer $\mathcal{P}_s$, language model LM, pre-net $h^{\text{pre}}$, and post-net $h^{\text{post}}$).

#### 3.2.1 SPEECH RECOGNITION AND TRANSCRIPT CONTINUATION

The first loss term is a combination of a speech recognition loss $\mathcal{L}_{\text{ASR}}$ and a transcript continuation loss $\mathcal{L}_{\text{LM}}$, which are given by:

$$\mathcal{L}_{\text{ASR}}(y_p, \hat{y}_p) = \text{CE}(y_p, \hat{y}_p), \quad \mathcal{L}_{\text{LM}}(y_c, \hat{y}_c) = \text{CE}(y_c, \hat{y}_c), \tag{5}$$

where CE denotes cross-entropy, which quantifies the dissimilarity between the predicted distribution over $\hat{y}_p, \hat{y}_c$, and the corresponding ground truth distribution over $y_p, y_c$. This objective increases the likelihood of the text $[y_p, y_c]$ under the conditional distribution modeled by the LM.

#### 3.2.2 SPEECH CONTINUATION

The speech continuation objective is formulated as a regression task, predicting each frame's spectrogram channels independently given previous frame spectrogram predictions and the ASR and LM context. To promote convergence and improve modeling power, we apply $\ell_1$ and $\ell_2$ regression losses on the spectrogram (Shen et al., 2020). These losses are applied to the feature-deltas of the spectrogram, and to the time-deltas of the spectrogram up to order $K$, giving "(discrete) derivative loss" terms. That is, for a tensor $z$ of dimension $T \times F$ we define:

$$\Delta_k^{\text{time}}(z) = z_{[1:T-k,:]} - z_{[k:T,:]}, \Delta_k^{\text{feat}}(z) = z_{[:,1:F-k]} - z_{[:,k:F]}, \tag{6}$$

$$\mathcal{L}_{1+2}(z, z') = ||z - z'||_1 + ||z - z'||_2^2. \tag{7}$$

For a ground truth spectrogram $x_c$ and the predicted spectrogram $\hat{x}_c$, the speech continuation loss is a combination of three objectives:

$$\mathcal{L}_{\text{s}}(x_c, \hat{x}_c) = \mathcal{L}_{1+2}(x_c, \hat{x}_c), \tag{8}$$

$$\mathcal{L}_{\text{f}}(x_c, \hat{x}_c) = \mathcal{L}_{1+2}(\Delta_1^{\text{feat}}(x_c), \Delta_1^{\text{feat}}(\hat{x}_c)), \tag{9}$$

$$\mathcal{L}_{\text{t}}(x_c, \hat{x}_c) = \sum_{k=1}^{K} \mathcal{L}_{1+2}(\Delta_k^{\text{time}}(x_c), \Delta_k^{\text{time}}(\hat{x}_c)) \tag{10}$$

The overall speech continuation loss is thus given by:

$$\mathcal{L}_{\text{Recon.}}(x_c, \hat{x}_c) = \mathcal{L}_{\text{s}}(x_c, \hat{x}_c) + \mathcal{L}_{\text{f}}(x_c, \hat{x}_c) + \mathcal{L}_{\text{t}}(x_c, \hat{x}_c). \tag{11}$$

#### 3.2.3 OVERALL LOSS

Using the above notation, our objective is:

$$\mathcal{L}_{\text{total}}(x, y) = \mathcal{L}_{\text{ASR}}(y_p, \hat{y}_p) + \mathcal{L}_{\text{LM}}(y_c, \hat{y}_c) + \mathcal{L}_{\text{Recon.}}(x_c, \hat{x}_c). \tag{12}$$

Since $\mathcal{L}_{\text{ASR}}$ and $\mathcal{L}_{\text{LM}}$ are cross-entropy losses and since $y = [y_p, y_c]$ (Eq.2), the overall speech recognition and transcript continuation loss can be written as:

$$\mathcal{L}_{\text{ASR}}(y_p, \hat{y}_p) + \mathcal{L}_{\text{LM}}(y_c, \hat{y}_c) = \text{CE}(y, \hat{y}) = \mathcal{L}_{\text{CE}}(y, \hat{y}) \tag{13}$$

where $\hat{y}$ is the concatenation of $\hat{y}_p$ and $\hat{y}_c$. This simplifies the overall loss to:

$$\mathcal{L}_{\text{total}}(x, y) = \mathcal{L}_{\text{CE}}(y, \hat{y}) + \lambda_r \mathcal{L}_{\text{Recon.}}(x_c, \hat{x}_c), \tag{14}$$

where $\lambda_r$ is a weighting coefficient. This simplification eliminates the necessity of the text-speech time alignment $\phi(s)$. Our approach can be seen as jointly optimizing three capabilities:

**Speech recognition** ($\mathcal{L}_{\text{ASR}}$): The combined model learns to transcribe speech audio into text. As we use a pre-trained speech encoder and a pre-trained language model, this objective encourages the alignment and integration of each model's functionality.

**Transcript continuation** ($\mathcal{L}_{\text{LM}}$): This reuses, maintains, and leverages the language model's ability to generate natural text as learned from its training scheme, for example, dialogue for a chat-optimized LM. Depending on the utterance, the decoder may further learn to use paralinguistic cues from the prompt speech to favor certain completions.

**Conditional speech synthesis** ($\mathcal{L}_{\text{Recon.}}$): We reuse the language model's autoregressive generation ability and direct it toward spectrogram reconstruction. As the teacher-forced transcript is available and the most "accessible" feature, the decoder learns to perform text-to-speech. In this way, the model can synthesize the LM's arbitrary textual continuations at inference time, including words not found in training. Finally, we expect that good spectrogram-level continuations require the preservation of speaker, prosody, and channel effects from the original speech prompt.

## 3.3 INFERENCE

In inference, the speech prompt $x_p$ is encoded by the speech encoder $\mathcal{E}$, then projected by $\mathcal{P}_s$ to the LM's dimension to give $x_p^{\text{lm}}$ (Eq. 3). Utilizing $x_p^{\text{lm}}$ and the start-of-sentence (sos) token, the language model decodes text in an autoregressive manner: $\hat{y} = \text{LM}([x_p^{\text{lm}}, \text{sos}])$ until eos is emitted, where $\hat{y}$ is a concatenation of the predicted transcript and continuation $[\hat{y}_p, \hat{y}_c]$. Following this, the language model decodes a spectrogram in an autoregressive manner. It predicts the next spectrogram feature estimate $\hat{x}_c(t)$ using prompt features $x_p^{\text{lm}}$, text prediction $\hat{y}$ and past estimated spectrogram features $\hat{x}_c(\leq t-1)$. Past spectrogram estimates $\hat{x}_c(\leq t-1)$ are projected to the language model dimension: $\hat{x}_c^p(\leq t-1) = h^{\text{pre}}(\hat{x}_c(\leq t-1))$. Then, $\hat{x}_c^p(t)$ is predicted at step $t$: $\hat{x}_c^p(t) = \text{LM}([x_p^{\text{lm}}, \text{sos}, \hat{y}, \hat{x}_c^p(\leq t-1)])$ The decoded output $\hat{x}_c^p(t)$ is then projected to the spectrogram domain using $h^{\text{post}}$: $\hat{x}_c(t) = h^{\text{post}}(\hat{x}_c^p(t))$. Finally, a vocoder converts the predicted spectrogram $\hat{x}_c$ into a waveform signal.

## 4 EXPERIMENTS AND RESULTS

### 4.1 DATA AND PRE-PROCESSING

To empirically evaluate the performance of the proposed approach, we conducted experiments on the Libri-Light dataset (Kahn et al., 2020). Libri-Light is a 60k hour English dataset consisting of unlabelled read speech from LibriVox audiobooks. For our training objective, the dataset was transcribed using a NST (Park et al., 2020) model trained on LibriSpeech (960 hours). We used a frozen neural vocoder, WaveFit (Koizumi et al., 2022), with its default hyperparameters to convert the predicted spectrograms into raw audio. Our proposed model was trained using 64 TPUv4 chips (Jouppi et al., 2023), over a duration of 48 hours. We give a comprehensive table of hyperparameters in Appendix A.1. We consider a predetermined set of 3-second prefixes denoted as $s = 3\text{sec}$. During training utterances with a length of less than 3 seconds are discarded. For Libri-Light, only $0.04\%$ of utterances are less than 3 seconds. To evaluate our model and the baseline models during testing, we utilize the test-clean test set from LibriSpeech (Panayotov et al., 2015). We employ the first 3 seconds of each utterance in the test set as a prompt to the models, excluding the ground truth transcripts. For semantic and acoustic quality, Spectron was trained with a LM of 350 million parameters, while for the question answering task, Spectron was trained with an LM of 1 billion parameters. The two models are identical except for the LM. In Sections 4.2.1 and 4.2.2 a 350M LM was used; in Section 4.2.3 a 1B LM was used.

### 4.2 BASELINES

We compare our method against existing spoken language models:

**GSLM:** We evaluate their best model, the HuBERT-L6 configuration with 200 token units, for conditional speech continuation. The model was trained on a filtered subset of Libri-Light (Rivière & Dupoux, 2021). **AudioLM:** We utilize the Libri-Light trained model described in their work. The two AudioLM models we compare against differ in the number of SoundStream residual vector quantizer (RVQ) layers they generate. One model generates the top 3 layers (**3-RVQ**), while the other model generates all 12 layers (**12-RVQ**). **TWIST:** We evaluate both the OPT-1.3B and LLaMA-7B-initialized versions of their models, which were trained towards quantized HuBERT representations.

Their models were trained on Libri-Light, Spotify podcasts (Clifton et al., 2020), The People's Speech (Galvez et al., 2021) and VoxPopuli (Wang et al., 2021). **SpeechGPT:** We evaluate their open-sourced model, which is based upon the LLaMA-7B model with HuBERT speech representations. This model is termed SpeechGPT-7B-com, and was trained using all 3 training stages in SpeechGPT. The model was trained using the Libri-Light and SpeechInstruct datasets.

### 4.2.1 SEMANTIC QUALITY

We employ the log-perplexity metric to evaluate the semantic quality of the speech output from the models. We use a state-of-the-art Conformer ASR system (Zhang et al., 2023b) trained on a proprietary English-only dataset to transcribe the speech continuation. Subsequently, we compute the log-perplexity of the predicted transcripts using GPT-2 medium (Radford et al., 2019) via the open-source `transformers` library (Wolf et al., 2020). The results presented in Table 1 demonstrate the performance gains of our method compared to previous approaches such as GSLM, where our method achieves an improvement of 170.91 in log-perplexity. Furthermore, when compared to the state-of-the-art AudioLM method, our approach outperforms both the 3-RVQ and 12-RVQ variants, exhibiting enhancements of 12.88 and 14.20 respectively. Moreover, the results in Table 1 reveal that our method exhibits improved performance compared to existing cascade methods.

### 4.2.2 ACOUSTIC QUALITY

We consider two metrics to capture acoustic quality and speaker consistency, respectively: **Naturalness Mean Opinion Score** (**N-MOS**; Nguyen et al., 2023)**:** This is reported solely for speech continuations. Human evaluators are tasked with assigning a rating on a five-point scale to denote the perceived naturalness of a given speech utterance, spanning from 1 (indicative of poor quality) to 5 (indicative of excellent quality). Tests were conducted using 20 randomly sampled utterances from the LibriSpeech test-clean test set. 30 raters participated in the tests. The prompts were not available to the raters. **Avg. speaker similarity:** We compute the speaker similarity between the input prompt and its generated continuation using the speaker encoder of the PnG-NAT TTS model (Morioka et al., 2022). We compute the speaker embeddings of both and measure the cosine similarity between each pair of embeddings. We report the average across the entire test set.

As seen in Table 2, our approach performs better than GSLM in terms of N-MOS with an improvement of 0.55 absolute. When compared to AudioLM, our approach is comparable to the 3-RVQ version and slightly inferior to the 12-RVQ version, with a decrease of 0.19 in N-MOS. One can see in Table 2 that the results of TWIST are similar to those of GSLM, and that Spectron outperforms the 1.3B and 7B versions by 0.4 and 0.65 respectively. SpeechGPT performs slightly inferior to Spectron, which outperforms it by a score of 0.3. Table 3 presents the results for average speaker similarity. Our method demonstrates a significant improvement of 0.31 over the GSLM method. When compared to AudioLM, our method outperforms both the 3-RVQ and 12-RVQ versions, with increases of 0.05 and 0.07 in average speaker similarity, respectively. Moreover, comparing to TWIST 1.3B and 7B, the proposed method improve the average speaker similarity by 0.18 and 0.19, respectively. These results indicate that comparable acoustic quality can be achieved with Spectron's simpler approach. Our model is trained end-to-end and utilizes the universal speech representation encoded by spectrograms. Note that SpeechGPT does not intend to preserve speaker identity, which is why its average speaker similarity is lower.

Table 1: Log-perplexity for completions of LibriSpeech utterances given a 3-second prompt. Lower is better.

| Method | Log-perplexity ($\downarrow$) |
| --- | --- |
| GSLM | 296.99 |
| AudioLM (3-RVQ) | 138.96 |
| AudioLM (12-RVQ) | 140.28 |
| TWIST (1.3B) | 229.53 |
| TWIST (7B) | 170.81 |
| SpeechGPT | 136.42 |
| Spectron (350M) | **126.08** |

Table 2: Naturalness Mean Opinion Score (N-MOS; Mean ± SE) for completions of LibriSpeech utterances.

| Method | N-MOS (↑) |
|---|---|
| GSLM | 3.13 ± 0.32 |
| AudioLM (3-RVQ) | 3.61 ± 0.29 |
| AudioLM (12-RVQ) | 3.87 ± 0.32 |
| TWIST (1.3B) | 3.28 ± 0.24 |
| TWIST (7B) | 3.03 ± 0.22 |
| SpeechGPT | 3.38 ± 0.30 |
| Spectron (350M) | 3.68 ± 0.29 |
| Ground Truth | 4.23 ± 0.33 |

Table 3: Average Speaker Similarity metric for completions of LibriSpeech utterances.

| Method | Speaker Sim. (↑) |
|---|---|
| GSLM | 0.11 |
| AudioLM (3-RVQ) | 0.37 |
| AudioLM (12-RVQ) | 0.35 |
| TWIST (1.3B) | 0.24 |
| TWIST (7B) | 0.23 |
| SpeechGPT | 0.05 |
| Spectron (350M) | **0.42** |

### 4.2.3 QUESTION ANSWERING

We propose examining whether the models can continue spoken sentences or questions with the appropriate answer. This can can be viewed as spoken generative QA; the correct answer must be produced out of infinite possible speech continuations. Note that except for SpeechGPT, all other methods (including ours) have not seen instruction data and are thus evaluated in a zero-shot fashion for spoken question answering. Given that the various spoken language models are evaluated with 3-second input contexts, we use TTS (via the publicly-available Google Cloud TTS service, voice en-US-Neural2-C) to synthesize questions that fit within this duration. The questions are drawn from an existing set and a new test set which we name LLaMA-Questions. **WebQuestions** (Berant et al., 2013) is an open-ended text QA NLP dataset. The dataset contains open-ended questions that are answerable via the Freebase database and are centered around a single named entity. **LLaMA-Questions** is an open-domain world knowledge QA dataset that we synthesized using LLaMA-7B. We prompted the model to provide questions and short answers regarding various topics. Overall, we gathered 300 questions in this manner and generally verified the answers. **Answer accuracy:** We use a Conformer ASR system (Zhang et al., 2023b) to transcribe the answers of the models. If the text answer is contained in the transcript, we count the answer as being correct (as zero-shot models are merely continuing the prefix audio).

The results presented in Table 4 demonstrate the performance of the proposed model in comparison to other existing models. Specifically, the proposed model exhibits an accuracy of 22.9% on LLaMA-Questions, while SpeechGPT achieves a comparable accuracy of 21.9%. Note that in contrast to SpeechGPT's utilization of a larger model architecture comprising 7 billion parameters, our proposed method uses a more modest 1 billion parameter LM for comparable results. In contrast, TWIST models with 1.3 billion and 7 billion parameters demonstrate lower accuracies of 1% and 0.5% respectively. Upon careful examination, it becomes evident that these models predominantly generate completions of input questions rather than providing substantive answers. AudioLM 3-RVQ, AudioLM 12-RVQ and GSLM achieved accuracy of 7%, 6.7% and 4%, respectability, which is likely due to the fact that the underlying Transformer architecture is not pre-trained on a large language model. Similarly, on the Web Questions test set, the proposed model attains an accuracy of 6.1%, while SpeechGPT yields a comparable accuracy of 6.5%. Again, TWIST models with 1.3 billion and 7 billion parameters achieve accuracies of 0.7% and 1.1% respectively, further reinforcing the observed trend of completion-centric behavior rather than direct question answering. Additionally, models such as AudioLM 3-RVQ, AudioLM 12-RVQ, and GSLM exhibit accuracies of 2.3%, 2.3%, and 1.5% respectively, which can likely be attributed to the absence of pre-training on a large-scale language model within the underlying Transformer architecture.

Audio samples and our spoken QA dataset can be found on the project website.

### 4.2.4 ABLATION ANALYSIS

To understand the individual impacts of various components within the proposed approach, an ablation study was conducted. We measure the log-perplexity over the test-clean test set of the LibriSpeech

Table 4: Accuracy (%) on spoken question answering datasets.

| Method | Web Questions (↑) | LLaMA-Questions (↑) | Zero-Shot |
|--------|-------------------|---------------------|-----------|
| GSLM | 1.5 | 4.0 | ✓ |
| AudioLM (3-RVQ) | 2.3 | 7.0 | ✓ |
| AudioLM (12-RVQ) | 2.3 | 6.7 | ✓ |
| TWIST (1.3B) | 0.7 | 1.0 | ✓ |
| TWIST (7B) | 1.1 | 0.5 | ✓ |
| SpeechGPT (7B) | 6.5 | 21.9 | × |
| Spectron (1B) | 6.1 | 22.9 | ✓ |

dataset (Panayotov et al., 2015). This study involved removing each specific component in isolation. (i) Disabled intermediate loss on text ("-$\mathcal{L}_{CE}$") (ii) removed spectrogram derivative loss ("-($\mathcal{L}_f + \mathcal{L}_t$)") (iii) removed pre-training of the language model LM, letting it train from scratch (iv) removed pre-training of the speech encoder $\mathcal{E}$ and training it from scratch (v) removed pre-training of both the speech encoder $\mathcal{E}$ and language model LM, training the entire model from scratch. The findings are summarized in Table 5. The results demonstrate that each of the aforementioned components contributes to the overall performance enhancement of the proposed approach. Notably, the ASR & LM cross-entropy loss $\mathcal{L}_{CE}$ and the spectrogram derivative loss $\mathcal{L}_f + \mathcal{L}_t$ have the most significant impact, leading to a degradation of 661.81 and 588.35 in the log-perplexity score, respectively. Furthermore, the incorporation of the pre-trained speech encoder and pre-trained language model exhibits a discernible decline in performance, resulting in a degradation of 87.17 and 75.63 in the log-perplexity score, respectively. Notably, when both the speech encoder and pre-trained language model are removed, a degradation of 118.31 in the log-perplexity score is observed.

Table 5: Ablation analysis.

| Model | Log-perplexity (↓) |
|-------|--------------------|
| Proposed Spectron (350M) | 126.08 |
| $-\mathcal{L}_{CE}$ | 714.43 |
| $-(\mathcal{L}_f + \mathcal{L}_t)$ | 787.89 |
| $-$ Pre-trained LM | 201.71 |
| $-$ Pre-trained speech encoder | 213.25 |
| $-$ Pre-trained LM & speech encoder | 244.39 |

## 5 LIMITATIONS AND FUTURE WORK

The limitation of our work is the high time and space complexity of generating spectrogram frames. Since spectrogram frames are computed with a rate of 12.5 ms, generation of long speech utterances is not possible. We hypothesize that potential solutions include generating multiple spectrogram frames from each hidden representation. Another limitation is that text and spectrogram decoding processes are not parallelizable. This hinders the ability to use Spectron in streaming scenarios and introduces a small latency between audio input and output. We leave the development of a parallelized decoding algorithm for future work. We further recognize that biases in the pre-trained language model may be sustained in our model, we refer to Google (2023) for a detailed discussion of ethical considerations for text-based language models.

## 6 CONCLUSION

We proposed Spectron, a neural direct speech continuation model that can be trained end-to-end and operates in the spectrogram domain. We showed that a pre-trained language model can be given speech recognition and generation capabilities post-hoc, by fine-tuning on continuation tasks using a pre-trained speech encoder and a novel training objective. The result is a model that benefits from the pre-training of both models and outperforms previous spoken language models on various metrics.

ACKNOWLEDGMENTS

The authors would like to thank Heiga Zen, Neil Zeghidour, Eugene Kharitonov, Tal Schuster, Bryan Richter, Christian Frank, Marco Tagliasacchi, Nadav Bar, and the rest of the Google Research team for helpful discussions and previous work on data preparation. The contribution of Alon Levkovitch is part of his Ph.D. thesis research conducted at Tel-Aviv University.

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

## A  APPENDIX

### A.1  TABLE OF HYPER-PARAMETERS

Table 6: Model hyper-parameters used in the experiments. ("$\times n$": $n$ layers)

| | |
|---|---|
| *Input & Output* | |
| Sample rate (Hz) | 16,000 |
| Mel channels | 128 |
| Mel lower band (Hz) | 20 |
| Mel upper band (Hz) | 8,000 |
| Frame size (ms) | 50.0 |
| Frame step (ms) | 12.5 |
| *SpecAugment* | |
| Freq blocks | 2 |
| Time blocks | 10 |
| Freq mask max bins | 27 |
| Time mask max frames | 40 |
| Time block max length ratio | 0.05 |
| *Speech Encoder* | |
| Conformer dims | 1024 |
| Attention heads | 8 |
| Conv kernal size | (3, 3) |
| Conv stride size | (2, 2) |
| *Language Model* | |
| Transformer (dim $\times$ layers) | 1024 |
| Dim per head | 64 |
| Hidden dims | 4096 |
| Num heads | 16 |
| Vocab size | 256,000 |
| *WaveFit vocoder* | |
| Iterations | 5 |
| UBlock upsampling factors | [5, 5, 2, 2, 2] |
| STFT loss resolutions | 3 |
| Hann win size, frame shift, FFT size res 1 | [160, 32, 512] |
| Hann win size, frame shift, FFT size res 2 | [400, 80, 1024] |
| Hann win size, frame shift, FFT size res 3 | [800, 160, 2048] |
| Multi-period discriminator | Kong et al. (2020) |
| Multi-period discriminator loss weight | 1.0 |
| *Training* | |
| Optimizer | Adam (Kingma & Ba, 2015) |
| Learning rate schedule | Vaswani et al. (2017) |
| Learning rate (peak) | $3.5 \times 10^{-4}$ |
| Warm-up steps | 8K |
| Batch size | 128 |
| Continuation loss weight $\lambda_r$ | 0.1 |
| Derivative loss order $K$ | 3 |

### A.2  EXTENDED COMPARISON TO PREVIOUS METHODS

Given the variation in pre-trained models across the methods discussed in this paper, we find it fit to conduct a more comprehensive comparison of their performance. This detailed comparison is presented in Table 7. Regarding the Word Error Rate (WER) assessment for ASR systems, the WER scores are sourced from the SUPERB benchmark paper (Yang et al., 2021). WERs are reported on the LibriSpeech test-clean test set. It's important to note that these scores rely solely on the speech encoder type due to limited data availability for all utilized models. For instance, models such as mHuBERT (Lee et al., 2022) employed in SpeechGPT and the New Frequency HuBert adapted and trained within TWIST exist solely as tokenization models and lack dedicated ASR model forms.

The performance comparison of speech encoders referenced in various methods within this paper is depicted in Table 7. Notably, the performance of the speech encoders on the LibriSpeech test set is comparable. However, concerning the language models (LMs) utilized, more variation is evident among the methods. LMs span a spectrum, ranging from larger models such as SpeechGPT and TWIST employing 7B LMs, to intermediate-sized models like Spectron and AudioLM employing approximately 1B LMs, and finally, GSLM utilizing a smaller 200M parameter LM. It is widely acknowledged that LM performance is significantly influenced by model size (Kaplan et al., 2020). Moreover, diverse datasets have been employed across these systems, including LibriSpeech, Libri-Light, SpeechInstruct, VoxPopuli (Wang et al., 2021), Common-Voice (Ardila et al., 2020), Spotify (Clifton et al., 2020), and The People's Speech (People; Galvez et al., 2021).

Table 7: Comparison of different models mentioned in this paper.

| Detail | Spectron | SpeechGPT | TWIST | GSLM | AudioLM |
|---|---|---|---|---|---|
| LM #Params | 1B / 350M | 7B | 1B/7B | ∼200M | 0.9B |
| LM Type | PaLM-2 | LLaMA | OPT/LLaMA | None | None |
| Speech Encoder | USM | mHuBERT | T-HuBERT | HuBERT | Wav2Vec |
| Speech Encoder #Params | 600M | 317M | 317M | 317M | 600M |
| WER of Speech Encoder | 3.1 | 2.94 | 2.94 | 2.94 | 3.1 |
| Speech Encoder Dataset | Web-scale LibriSpeech | VoxPopuli 100k | LibriSpeech VoxPopuli CommonVoice Spotify Fisher | LibriSpeech Libri-Light | Libri-Light |
| Training Dataset | Libri-Light | LibriSpeech Speech Instruct | LibriSpeech Spotify People VoxPopuli | Libri-Light | Libri-Light |
| #Training Examples | 60k | 60k + 38k | 150k | 60k | 60k |

