# OpenReview forum: "Spoken Question Answering and Speech Continuation Using Spectrogram-Powered LLM"
_ICLR.cc/2024/Conference — ICLR 2024 poster_

### Official Review · Reviewer_Qrsa · 2023-10-24

**Soundness:** 3 good
**Presentation:** 4 excellent
**Contribution:** 3 good
**Rating:** 8
**Confidence:** 4

**Summary:**

In this paper, the authors leverage a pretrained Language Model (LLM) and a pretrained speech encoder to tackle spoken question answering and speech continuation tasks. The proposed model processes spectrograms directly as inputs and outputs while utilizing a novel end-to-end objective for training. This objective implicitly supervises speech recognition, text continuation, and conditional speech synthesis tasks. Comprehensive experimental results validate the semantic and acoustic quality of the proposed model, showcasing its efficacy.

**Strengths:**

1. This work introduces a new end-to-end training paradigm for spoken language modeling, efficiently employing the pretrained Language Model (LLM) to enhance semantic quality.
2. The experimental results, along with the audio samples in the supplementary material, confirm the semantic and acoustic performance of the model.
3. The availability of the released test set offers the research community a valuable benchmark for assessing the semantic quality of spoken language models.
4. The presentation is clear and easy to understand.

**Weaknesses:**

1. The association between generated text and speech can be further evaluated. For instance, performing a Word Error Rate (WER) test using speech, [xp, xc], and transcriptions [yp, yc] would be advantageous. In addition, replacing the text with a sequence of null tokens of the same length can help showcase the impact of text on semantic quality. These analyses would lead to a better understanding of the role that text plays in the proposed model.

**Questions:**

1. Could you please provide additional information about the training and inference in the Spoken QA task? Does SPECTRON undergo training for the spoken QA task, whereas the baselines, such as AudioLM and GSLM, does not?

2. In the provided Spoken QA demos, SpeechGPT tends to generate lengthier and more detailed answers, while SPECTRON prefers more concise responses. Could the comparable performance of both models be attributed to the nature of the QA task, such as the prevalence of shorter answers in the test set?

**Details Of Ethics Concerns:**

None.

---

> ### Author Response · Authors · 2023-11-15
> **Response to Reviewer Qrsa**
>
> Thank you for your detailed comment and constructive suggestions. We appreciate the time and effort you have invested in reviewing our submission.
>
> **Weaknesses:**
>
> **The association between generated text and speech can be further evaluated.**
>
> When obtaining predicted transcription and continuation y_hat, we cannot split it to the prompt transcription and the continuation transcription. For the predicted text sequence, we do not have a way to know where the switch between transcription and continuation happens. Thus, we cannot compute WER against prompt only or against continuation only. However we measured the ASR performance of the proposed model trained exclusively for ASR without speech continuation, the results were a WER of 3.1 / 4.9 on Librispeech test-clean / test-other.
>
> **In addition, replacing the text with a sequence of null tokens of the same length can help showcase the impact of text on semantic quality.**
>
> We have performed this analysis, we replaced the text with a sequence of null tokens, with a sequence of characters (for example “a a a a”) and with some other arbitrary token. We see consistent results, that when doing this the speech continuation of the model contains mumbling or long human voices. We can thus conclude that indeed the text has an impact on the semantic quality of the generated speech.
>
> **Questions:**
>
> **Could you please provide additional information about the training and inference in the Spoken QA task? Does Spectron undergo training for the spoken QA task, whereas the baselines, such as AudioLM and GSLM, does not?**
>
> No specific training has been done on spoken question answering. Spectron was only trained for speech continuation as described in Section 3.2. AudioLM, GSLM and TWIST also haven't been explicitly trained for spoken question answering. SpeechGPT, on the other hand, was trained using a carefully curated multimodal question answering dataset. We have added a column to Table 4.
>
> **In the provided Spoken QA demos, SpeechGPT tends to generate lengthier and more detailed answers, while SPECTRON prefers more concise responses. Could the comparable performance of both models be attributed to the nature of the QA task, such as the prevalence of shorter answers in the test set?**
>
> Spectron hasn't been trained using spoken QA datasets, we only used those datasets for evaluation. SpeechGPT was explicitly trained for question answering using carefully curated spoken QA dataset, therefore its answers are more detailed. Also, note that the evaluation strategy of the spoken QA datasets does prefer short answers. We define that an answer is correct by checking if the correct answer is contained in the model output. For example, if the model output is “The capital of France is Paris” and the correct answer is “Paris” we say that the model answered the question correctly. Thus, long and detailed answers have a higher chance of containing the correct answer inside of their response.

---

> > ### Comment · Reviewer_Qrsa · 2023-11-22
> >
> > Thanks for addressing my concern. I will keep my original score.

---

> > > ### Author Response · Authors · 2023-11-22
> > > **Response to Official Comment by Reviewer Qrsa**
> > >
> > > Thank you once again for your time and insights.

---

### Official Review · Reviewer_zUkq · 2023-10-30

**Soundness:** 4 excellent
**Presentation:** 4 excellent
**Contribution:** 3 good
**Rating:** 8
**Confidence:** 3

**Summary:**

This paper proposes a novel approach to adapt the pre-trained LLMs for spoken question answering and speech continuation. Specifically, a pre-trained speech model as well as some projection layers such as connector layer, pre-net, and post-net are utilized to process speech signals in the same way as text tokens in the LLMs. In this way, the adapted LLMs can take speech inputs and generate speech outputs and train speech recognition, text continuation, and speech synthesis using only paired speech-text pairs in an end-to-end fashion, where the text transcripts can be treated as a cross-modal chain-of-thought. The proposed spectrogram-powered LLMs surpasses existing spoken language models in speaker preservation, semantic coherence, as well as spoken question answering.

**Strengths:**

**Originality:** The concepts and methods of Spectron is novel. It adapts exiting pre-trained LLMs for spoken language modeling, and trains an end-to-end system that takes speech inputs and generates speech outputs.

**Quality:** This paper has solid experimental results. The proposed system doesn't achieve state-of-the-art results in some metrics though.

**Clarity:** This paper is well-organized and well-written.

**Significance:** The impact of the paper is good. However, the scale of the proposed system is relatively small compared to OpenAI voice-version ChatGPT.

**Weaknesses:**

I don't see major weaknesses. It would be great if the authors can also compare their system with OpenAI voice-version ChatGPT (https://openai.com/blog/chatgpt-can-now-see-hear-and-speak).

**Questions:**

Will Spectron still work when using a LLM that has 100B parameters? Currently, Spectron uses 1B language model (PaLM 2), and that is much smaller than state-of-the-art LLMs. In addition, what happens if you use Whisper (https://cdn.openai.com/papers/whisper.pdf) to transcribe Libri-Light and obtain training transcripts $y$ ?

---

> ### Author Response · Authors · 2023-11-15
> **Response to Reviewer zUkq**
>
> Thank you for your detailed comment and constructive suggestions. We appreciate the time and effort you have invested in reviewing our submission.
>
> **Weaknesses:**
>
> **I don't see major weaknesses. It would be great if the authors can also compare their system with OpenAI voice-version ChatGPT (https://openai.com/blog/chatgpt-can-now-see-hear-and-speak).**
>
> Thank you for the suggestion. It's important to note that the voice-version of ChatGPT was released post the abstract submission deadline and in close proximity to the final paper submission. Additionally, the detailed model architecture, performance metrics, and training methodology of the voice-version ChatGPT have not been officially published. Hence, directly comparing to the voice-enabled version of ChatGPT is not straightforward, despite its impressive capabilities. Nonetheless, we believe that the comparison between these two systems holds significant interest and should be addressed in future work.
>
> **Questions:**
>
> **Will Spectron still work when using a LLM that has 100B parameters? Currently, Spectron uses 1B language model (PaLM 2), and that is much smaller than state-of-the-art LLMs.**
>
> We experimented with a larger LLM, however due to computational constraints we were not able to complete the training process. We think that Spectron will also work well with a larger LLM, though more data or parameter freezing may be needed to avoid overfitting.
>
> **In addition, what happens if you use Whisper (https://cdn.openai.com/papers/whisper.pdf) to transcribe Libri-Light and obtain training transcripts y?**
>
> Whisper obtains a 2.7 / 5.2 error rate on Librispeech test-clean/test-other. The model used for the transcription of the Librilight dataset is the NST model. It achieves a WER of 1.7/3.4 on Librispeech test-clean/test-other datasets. We thus believe that using the Whisper model to transcribe that dataset shouldn’t have a significant effect on the performance of the system.

---

### Official Review · Reviewer_hbZt · 2023-10-31

**Soundness:** 3 good
**Presentation:** 3 good
**Contribution:** 3 good
**Rating:** 6
**Confidence:** 4

**Summary:**

This paper introduces a new spoken language model that operates on spectrograms and can simultaneously perform speech recognition, speech continuation, and text continuation (auto-regressive language modeling). The authors propose a model with a speech encoder, an LLM decoder, and a post network that takes in a sequence of speech input to produce its transcription, textual completion, and speech completion.  Experiments on LibriLight show that the proposed Spectron obtains better acoustic and semantic quality compared to other work. They also extend the approach to Spoken Question Answering and show that Spectron can do this almost as well as SpeechGPT.

**Strengths:**

1. The proposed approach is clear and simple. Experimental results demonstrate benefits from this approach compared to others.

2. The idea of prompting order imposed that the authors claim resembles CoT is interesting.

**Weaknesses:**

1. The MOS evaluation of synthesis is bereft of details - it is important to mention the number of raters and the number of examples rated. The lack of information makes it hard to contextualize reported MOS scores. Further, since this paper is evaluating speech continuation as opposed to text-to-speech synthesis (TTS), it is unclear if standard MOS is the best evaluation strategy. Do the raters hear the original prompt before hearing the continuation from the model? Does the MOS in this case measure naturalness of the continuation or something else ?

2. The CoT idea is interesting as I mentioned before, but this paper could benefit from some analysis on how inter-dependent these outputs are in the trained model. That is, if the transcription contains errors, does that necessarily lead to poor text or speech continuation?

**Questions:**

A. No details are provided on MOS tests - could the authors share some details ?

B. What was the WER of the Librispeech model used to obtain pseudo-labels on LibriLight ? Do the authors have any idea about label quality ?

C. The paper says "we use TTS to synthesize questions that fit within this duration" - what does this mean? Do you truncate long questions or modify the durations of phones/words to fit within 3s? Or do you drop questions that are longer than 3s?

---

> ### Author Response · Authors · 2023-11-15
> **Response to Reviewer hbZt**
>
> Thank you for your detailed comment and constructive suggestions. We appreciate the time and effort you have invested in reviewing our submission.
>
> **Weaknesses:**
>
> **The MOS evaluation of synthesis is bereft of details**
>
> In the MOS experiments 30 raters rated 20 randomly sampled examples. We employed MOS experiments to gauge the naturalness of the synthesized continuous speech. raters were not exposed to the initial prompt. To further clarify this issue, we changed the name of this metric in the revisioned paper to N-MOS (N for naturalness). We report Naturalness MOS as only one of many axes of evaluation; e.g. we transcribed the resulting speech and measured both perplexity and accuracy using our methods.
>
> **The CoT idea is interesting as I mentioned before**
>
> We conducted experiments involving the introduction of errors into the ASR transcription and text continuation to examine the interconnectedness of the decoding process. Our observations confirm a clear interdependence among the outputs. Specifically, introducing errors into both the text continuation and ASR transcription resulted in notably inferior speech generation outcomes. Consequently, we conclude that the cohesion of the entire process significantly influences the model's performance. We believe that this issue warrants further attention in future research.
>
> **Questions:**
>
> **No details are provided on MOS tests - could the authors share some details ?**
>
> In the MOS experiments 30 raters rated 20 randomly sampled examples. We use the MOS experiments to measure the naturalness of the synthesized continuation speech without the prompt. We have added these details to the revised version.
>
> **What was the WER of the Librispeech model used to obtain pseudo-labels on LibriLight ? Do the authors have any idea about label quality ?**
>
> The Librispeech model that was used to obtain the pseudo-labels is the NST model. It achieves a WER of 1.7/3.4 on Librispeech test-clean/test-other datasets. From manually sampling some of the training data, it seems that the quality of the pseudo labels is high.
>
> **The paper says "we use TTS to synthesize questions that fit within this duration" - what does this mean? Do you truncate long questions or modify the durations of phones/words to fit within 3s? Or do you drop questions that are longer than 3s?**
>
> We drop questions that are longer than 3s. Since previous methods (GSLM and AudioLM) use only 3s prompts, we only considered questions whose synthesized spoken version was shorter than 3s.

---

> ### Comment · Reviewer_hbZt · 2023-11-22
>
> I thank the authors for their response, and for addressing the questions and weaknesses.
>
> 1. I still believe that the manner in which MOS was used here is not ideal -> raters should have been exposed to the initial prompt and asked to rate how natural the continuation was as opposed to just the audio.
>
> 2. 20 samples are likely small for such tests.
>
> 3. The response to CoT is appreciated, but some analysis and numbers in the paper would substantiate the author's motivation behind choosing this CoT style of input.

---

> > ### Author Response · Authors · 2023-11-22
> > **Response to Official Comment by Reviewer hbZt**
> >
> > We thank the reviewer for the comprehensive feedback.
> >
> > 1. We have performed a N-MOS test where the raters were also exposed to the prompt before evaluating the continuation. The results were 3.69±0.40 N-MOS, which is similar to the original score.
> >
> > 2. We repeated the N-MOS test for Spectron, this time with 50 examples. The results, 3.66±0.29, align closely with those from the initial 20-example test.
> >
> > 3. We ran a TTS experiment where we utilized the prompt for acoustic conditioning and manipulated the text in the CoT to match the requested text. The N-MOS results for this experiment are 3.40±0.31, which is a fair outcome, considering that the system was not specifically trained for TTS.
> >
> > 4.  We are currently conducting an experiment to measure the impact of transcription errors on text and speech continuation. Unfortunately, as we initiated the experiment upon reading your comment the author's response phase is nearing its end. We intend to provide comprehensive updates in the final version of the paper.

---

> > > ### Comment · Reviewer_hbZt · 2023-11-22
> > >
> > > I thank the authors for their response.

---

### Official Review · Reviewer_6UiY · 2023-11-03

**Soundness:** 3 good
**Presentation:** 2 fair
**Contribution:** 2 fair
**Rating:** 5
**Confidence:** 4

**Summary:**

This paper proposes a new training scheme to build a large speech and language model. The idea is to split the paired speech utterance into the first 3-second segment and the rest. The first task is to predict the corresponding sentence given the first 3-second segment. This corresponds to 1) performing ASR corresponding to the first 3-second audio segment and 2) predicting the rest of the sentence conditioned by the first 3-second audio segment and its transcriptions, corresponding to the language modeling task if we ignore the first 3-second audio segment. Note that the actual implementation does not require the split in the text part. The second task is to perform the speech continuation task of the rest of the audio segment conditioned on the first 3-second audio segment and previously estimated sentence. Thus, this training scheme holds ASR, text continuation, and speech continuation tasks with a single training framework with the standard paired speech utterances. This model is built upon various speech and text pre-trained models and is fine-tuned with the public speech database based on Libri-Light. The paper also has multiple comparisons with the other methods for speech continuation and spoken QA tasks and shows the superiority of the proposed method.

**Strengths:**

- Building a speech foundation model by leveraging an LLM is a hot topic in ML and AI. Also, providing more powerful understanding capability for speech models is desired.
- A novel training algorithm to (implicitly) perform ASR, text continuation, and speech continuation tasks with the standard paired speech utterances.
- The paper shows a strong performance compared with other speech LM methods.

**Weaknesses:**

- The paper lacks the reproducibility and accessibility of the results due to several pre-training models, which are not publicly available or cannot be reproduced due to the inaccessible training data (e.g., WaveFit and state-of-the-art Conformer ASR system (Zhang et al., 2023b)). I appreciate your efforts in mitigating the issue (e.g., the use of Libriright and the release of the SQA test set), but I think the paper still has this issue.
- Due to the above issue, it is not clear whether the superiority of the proposed method compared with other speech LMs comes from their novel training schemes or strong pre-trained models. Thus, the effectiveness, especially for the comparisons with other speech LMs, is weak.
- Clarity: Section 2 requires more improvements. There are many different aspects between the related studies and the proposed method, and it is not clear what is the advantage of this method. I recommend you rewrite Section 2 to categorize the different aspects (e.g., model architecture, training method, pre-training data, etc.) and emphasize the distinction between the proposed method and others.

**Questions:**

- Can you evaluate the ASR performance of this method? It would probably not be nice due to the over-prediction function in the ASR prediction phase (it will predict more than what was spoken), but it would be an interesting result to report.
  - I have the same question for TTS. In this case, due to the lack of conditions, it may not work at all (?).
- Why is it 3 seconds? Would it robustly work even if we throw more than (or less than) 3-second spoken prompts?
- Similarly, how did you deal with sentences that are less than 3 seconds during training? Can you discard them or concatenate neighboring sentences to make it longer than 3 seconds?
- How about combining multiple sentences and using the original first sentence as a chunk instead of a 3-second chunk? In this case, we would also have access to the transcription corresponding to the spoken prompt part easily. We can have more precise control of the text output to perform ASR inside the framework explicitly.
- Around equation (3), $\mathcal{P}_s$: why does $\mathcal{P}_s$ depend on position $s$?
- In section 4.1, the last sentence, "For semantic and acoustic quality, Spectron was trained with a model of 350 million parameters, while for the question answering task, Spectron was trained with a model of 1 billion parameters," I could not understand it. Do you mean you use 350M models for Sections 4.2.1 and 4.2.2 and 1 B models for Section 4.2.3? How are they different?
- Is the question-answering task performed with zero-shot without fine-tuning it? Please clarify it.

---

> ### Author Response · Authors · 2023-11-15
> **Response to Reviewer 6UiY (Part 1 / 2)**
>
> Thank you for your detailed comment and constructive suggestions. We appreciate the time and effort you have invested in reviewing our submission.
>
> **Weaknesses**
>
> **The paper lacks the reproducibility and accessibility of the results**:
>
> The release of these upstream pre-trained models and code built atop them are beyond our control. We tried to mitigate this issue in other aspects; beyond training and (to-be released) test sets from public data, we also provide a full description of the model architectures, novel objectives and hyperparameters used in the training (Appendix A.1). With these, we believe our key contribution is reproducible with our test suite and the many public LLMs, ASR encoders, and vocoders (e.g., LLaMA, wav2vec2 CTC, HiFiGAN), especially as our method’s semantic improvements are a step-function over the other methods (except for the concurrent SpeechGPT work), and our acoustic improvements are a step-function over SpeechGPT. We welcome suggestions on how to further improve reproducibility for this discussion period and camera review.
>
> **Due to the above issue, it is not clear whether the superiority of the proposed method**:
>
> Thank you for the comment. In order to evaluate the origin of our models improvement we compare the pre-trained models that each method use:
>
> | Detail | Ours | SpeechGPT | TWIST | GSLM | AudioLM |
> |---|---|---|---|---|---|
> | LL # Params | 1B/350M | 7B | 1B/ 7B | ~200M  | 0.9B (0.3B per stage)  |
> | LM Type | Palm 2 | LLama | OPT (1B) / LLama (7B) | None | None |
> | Speech Encoder | USM | mHuBert  | T-Hubert | HuBert Large | Wav2Vec  |
> | Speech Encoder # Params | 600M | 317M | 317M | 317M | 600M |
> | WER of Speech Encoder | 3.1  | 2.94 | 2.94 | 2.94 | 3.1 |
> | Speech Encoder Dataset | Web-scale, Librispeech fine-tune | VoxPopuli 100k | Librispeech, VoxPopuli, Common Voic, Spotify, Fisher.  | Librispeech | Librilight |
> | Training Datasets | Librilight | Librilight, Speech Instruct | Librispeech, Librilight, Spotify podcasts, People dataset, VoxPopuli | Librilight | Librilight |
> | # Training Examples | 60k | 60k + 38k | 150k | 60k | 60k |
> | | | | |
>
> LLM pretrained model - Our method is based on the 1B parameters LLM whereas SpeechGPT and TWIST utilize a 7B parameters model. LLM performance is well known to be strongly influenced by the model size. Therefore, we do not believe that the language model used in our method is more powerful than the language models used in SpeechGPT or TWIST methods. In comparison with AudioLM, we use a comparable model size. We acknowledge that the GSLM model is smaller than our model.  Moreover, SpeechGPT is fine-tuned with cross-modal instruction objectives derived from carefully curated speech-text datasets. TWIST must learn its speech capability from scratch, and so they use approximately 150k hours of speech-text datasets. Our method exhibits competitive results for multiple tasks using only the public Libri-Light dataset (60k hours) for adaptation.
>
> ASR / Tokenization - TWIST, SpeechGPT and GSLM all use the HuBERT model for tokenization. On the ASR task, from the SUPERB paper, which uses the same type of representation, the Hubert-Large model achieves a WER of 2.94 on the LibriSpeech test-clean test set. In comparison, the USM model we used has a performance of 3.1 on the test-clean test sets. Thus, we do not think that the speech-encoder we have used is more powerful than the open-source methods or any of the encoders used in the methods we have compared against.
>
> **Clarity: Section 2 requires more improvements.**
>
> Thank you for the comment. We re-wrote Section 2 based on your suggestions (e.g., model architecture, training method, pre-training data, etc.). The re-written section can be viewed in the updated version of the paper.  Additionally, extra information about the related work can be found in Appendix A.2.

---

> ### Author Response · Authors · 2023-11-15
> **Response to Reviewer 6UiY (Part 2 / 2)**
>
> **Questions:**
>
> **Can you evaluate the ASR performance of this method?**
>
> For the ASR task, we have computed the WER between the generated text and the original labels. As you state, it is important to note that the text also contains the continuation, and thus, the WER is not ideal. A large portion of the error comes from the continuation since there is a large variance in text continuation. When comparing a text continuation to a transcript, even a deviation of a single token will mean that the whole sequence that comes after the error is wrong with respect to the ground truth transcription. However, we measured the ASR performance of the proposed model trained exclusively for ASR without speech continuation, the results were a WER of 3.1 / 4.9 on Librispeech test-clean / test-other. For TTS, we are currently exploring the transformation of Spectron into TTS and aim to provide an update shortly.
>
> **Why is it 3 seconds? Would it robustly work even if we throw more than (or less than) 3-second spoken prompts?**
>
> In order to be consistent with the previous work (GSLM and AudioLM), we use prompts of 3 seconds or less during testing.
>
> **Similarly, how did you deal with sentences that are less than 3 seconds during training? Can you discard them or concatenate neighboring sentences to make it longer than 3 seconds?**
>
> We discard sentences shorter than 3 seconds during training. Note that in the LibriLight and Librispeech the amount of examples that are less than 3 seconds are only 0.04% and 2.73% respectively. Thus, discarding these examples is a minor change to the datasets.
>
> **How about combining multiple sentences and using the original first sentence as a chunk instead of a 3-second chunk? In this case, we would also have access to the transcription corresponding to the spoken prompt part easily. We can have more precise control of the text output to perform ASR inside the framework explicitly.**
>
> Thanks for the suggestion. We focused on continuations of 3-second prefixes to match past work, but we can try this in future work regarding the use of the Spectron framework as an ASR or TTS system.
>
> **Around equation (3), $\mathcal{P}_s$: why does $\mathcal{P}_s$ depend on position $s$?**
>
> Thanks for finding this typo, we removed the $s$ from the projection layer $\mathcal{P}$.
>
> **In section 4.1, the last sentence, "For semantic and acoustic quality, Spectron was trained with a model of 350 million parameters, while for the question answering task, Spectron was trained with a model of 1 billion parameters," I could not understand it. Do you mean you use 350M models for Sections 4.2.1 and 4.2.2 and 1B models for Section 4.2.3? How are they different?**
>
> Thank you for the comment. Yes, we mean that we use the 350M LM for Sections 4.2.1 and 4.2.2 and 1B LM model for Section 4.2.3. All other configurations of the models are identical in both methods. We did this to better compare Spectron in the audio continuation task since other audio continuation methods use smaller models. We wrote it explicitly in the revision of the paper.
>
> **Is the question-answering task performed with zero-shot without fine-tuning it? Please clarify it.**
>
> The question answering task is performed without fine-tuning; it is performed zero-shot. Spectron was only trained for speech continuation as described in Section 3.2. The world knowledge encapsulated in the pre-trained LLM allows Spectron to correctly answer spoken questions.

---

> ### Author Response · Authors · 2023-11-22
> **Answer to Reviewer 6UiY Regarding TTS Experiment**
>
> We have successfully completed the TTS experiment for Spectron. In this experiment, we utilized the prompt for acoustic conditioning and manipulated the text in the CoT to match the requested text. The N-MOS results for this experiment are 3.40±0.31, which is a fair outcome considering that the system was not specifically trained for TTS.

---

### Author Response · Authors · 2023-11-15
**Overall Author Response**

We thank all the reviewers for their time and effort in reviewing our submission. We greatly appreciate your comments and suggestions and have responded to them individually. **We have also uploaded a new revision that improves our presentation as follows:**

To better clarify our contributions, we updated the text of section 2 and added a comparison table (Appendix A.2), following reviewer 6UiY’s request to better detail our differences and performance versus pre-existing methods/models referenced in this paper. We also restate our commitment to contribute LLamaQuestions back to the community.

To better document the soundness of our approach, in section 4 we have added the further experimental details requested by each reviewer.


We are happy to address further questions and concerns during this discussion period; let us know!

---

### Meta-Review · Area_Chair_Siyc · 2023-12-04

**Metareview:**

The paper introduces Spectron, a new spoken language model that directly processes spectrograms as both input and output. It leverages the audio capabilities of a pre-trained speech encoder using intermediate projection layers and adapts pre-trained LLMs for QA and speech continuation. It demonstrates competitive performance in speaker preservation and semantic coherence compared to other end-to-end spoken language models.

One of the four reviewers expressed concerns about the reproducibility, accessibility of the results, and clarity. The authors addressed these issues in their rebuttal, but there was no response from that reviewer. The remaining reviewers rated the paper as ‘accept’ or ‘marginally accept’ and acknowledged the authors’ rebuttal.

**Justification For Why Not Higher Score:**

The paper’s contributions do not appear to be significant enough, and the methods proposed lack sufficient novelty for a higher score.

**Justification For Why Not Lower Score:**

The test set that has been made available serves as an important benchmark for the research community to evaluate the semantic quality of spoken language models. The paper is presented in a manner that is both clear and comprehensible.

---

### Decision · Program_Chairs · 2024-01-16

Accept (poster)